# The joint effect of weight-adjusted waist index and physical activity on all-cause mortality in Chinese elderly patients with multimorbidity: A study based on the CLHLS from 2011 to 2018

**Bingbing Fan**[ID], **Kexin Ren**[ID]*, **Lang Li**

Physical Education of Jilin Normal University, Si ping City, Jilin Province, China

* jlsdrkx@163.com

## Abstract

### Background

The relationship between physical activity changes, weight-adjusted waist circumference index (WWI), and mortality risk among older Chinese adults with multimorbidity remains unclear. This study aimed to examine whether changes in physical activity and WWI modify the mortality risk by analyzing data from the Chinese Longitudinal Healthy Longevity Survey (CLHLS).

### Method

Our study was based on the 2011~2018 wave of the CLHLS, involving a study of 2,626 older adults with multimorbidity. Cox proportional hazards models were employed to estimate hazard ratios (HRs) and 95% confidence intervals (95% CIs) and a stratified analysis was conducted to assess the combined impact of WWI and physical exercise on all-cause mortality in patients with multimorbidity.

### Result

Patients with multimorbidity who engaged in regular physical activity exhibited a 41% reduction in all-cause mortality compared to those who had never been physically active (HR:0.59, 95% CI:0.49, 0.70). All-cause mortality was increased by 13% in patients with high WWI and multimorbidity compared to those with low WWI (HR:1.13, 95% CI:1.01, 1.27). Furthermore, WWI-stratified analyses revealed that varying physical activity profiles had a more pronounced protective or detrimental impact on all-cause mortality among multimorbidity patients in the high WWI group compared to the low WWI group.

**Editor:** Maha Gasmi, University of Manouba Higher Institute of Sports and Physical Education of Ksar Said: Universite de la Manouba Institut Superieur du Sport et de l'Education Physique de Ksar Said, TUNISIA

**Data availability statement:** FAN, Bingbing. CLHLS 2011-2018.dta. Ann Arbor, MI: Inter-university Consortium for Political and Social Research [distributor], 2025-05-06. https://doi.org/10.3886/E228681V1

**Funding:** The author(s) received no specific funding for this work.

**Competing interests:** The authors have declared that no competing interests exist.

## Conclusion

This study demonstrates that both initiating and maintaining physical activity significantly reduce mortality risk in multimorbid older adults, even those with higher WWI. Our findings support integrating structured exercise interventions and routine WWI monitoring into clinical care to improve survival outcomes in this population.

## 1. Introduction

Global population aging is accelerating at an unprecedented rate, with the number of individuals aged 65 and above expected to exceed one billion by 2030 [1]. This trend has led to many countries facing a growing proportion of elderly people. China currently has the largest elderly population, which is projected to reach approximately 402 million by 2040, accounting for about 28% of its total population [2]. The rise in multimorbidity among the elderly, driven by an aging population and lifestyle changes like physical inactivity and obesity, increases their risk of premature death compared to those with a single disease [3]. This issue presents a significant global challenge with wide-ranging implications for individuals, caregivers, and society.

The issue is particularly acute in low- and middle-income countries, which are estimated to bear 80% of the global burden of non-communicable diseases [4]. In response, China has restructured its healthcare delivery system to more effectively manage multimorbidity rather than duplicating services for individual non-communicable diseases [5]. It is important to note that, despite growing research on chronic diseases in China, the epidemiological pattern of multimorbidity remains unclear [6]. Furthermore, although there is compelling evidence that interventions can mitigate the impact of multimorbidity on patients, national reductions in its prevalence or burden remain imperceptible [3]. These findings underscore the need for more effective and widely accepted interventions aimed at mitigating the harms associated with multimorbidity.

The health benefits of physical activity are widely recognized in China. And an expanding body of research substantiates that physical activity is the most cost-effective intervention for the prevention of multimorbidity [7]. Similarly, evidence suggests that physical inactivity, particularly sedentary lifestyle habits, is a significant contributor to multimorbidity and mortality among older individuals [8]. This implies that physical activity is significant for patients with multimorbidity. However, it remains to be explored whether there are factors that may influence the relationship between physical activity and all-cause mortality in patients with multimorbidity.

Recently, emerging research suggests that the relationship between physical activity and human health may be influenced by weight-adjusted waist circumference (WWI) [9]. Most studies have shown that BMI cannot differentiate between fat and muscle mass, nor can it distinguish generalized obesity from central obesity [10]. Therefore, there is an urgent need for a more accurate indicator to assess the nuances of obesity and its potential impact on comorbidities. In contrast, the WWI combines waist circumference and body weight to provide a more precise measure

of body fat distribution [11]. A cohort study of 4,732 participants in Iran demonstrated the importance of the WWI and physical activity in reducing the risk of developing chronic diseases [12]. Higher levels of WWI have also been found to be associated with an increased risk of all-cause mortality and death due to chronic diseases [13]. Given the significant role of abdominal obesity in older adults' health and the limitations of BMI, WWI serve as a crucial tool for clarifying the association between obesity and chronic disease multimorbidity.

Nonetheless, there is a relative scarcity of studies examining the potential joint effect of WWI and varying physical activity profiles on all-cause mortality in patients with multimorbidity. Therefore, the objective of our study was to assess the relationship between changes in physical activity and WWI and the prevalence of co-morbidities and risk of death in Chinese older adults.

## 2. Materials and methods

### 2.1. Study population

This study analyzes data from the 2011–2018 wave of the China Longitudinal Healthy Longevity Survey (CLHLS), which provides complete mortality follow-up and all required covariates. The data comprise a nationally representative sample of individuals aged 65 and older across 23 provinces in China. Approval was granted by Peking University's Biomedical Ethics Committee (IRB00001052–13074), and written informed consent was obtained from all participants. Participants without multimorbidity prior to death, as well as those with missing data on physical activity, weight, waist circumference, or baseline covariates, were excluded, resulting in a final sample of 2,626 participants. The inclusion and exclusion criteria are detailed in Fig 1.

### 2.2. Weight-adjusted-waist index

During the investigation, weight (kg) was measured by measuring weighing scale without shoes and heavy clothes. For waist circumference measurement, participants were instructed to stand in an upright position with a calm exhalation. A

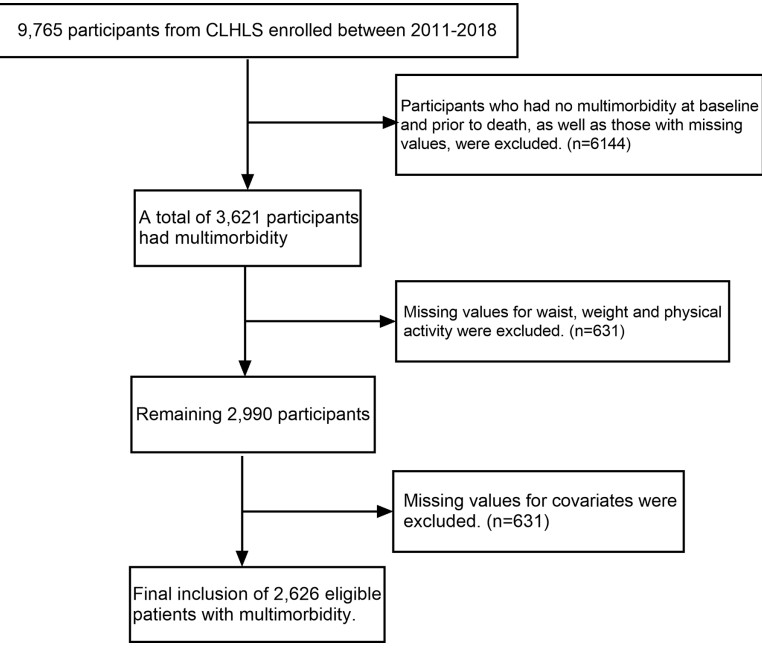

**Fig 1. Flowchart for Participant Screening and Exclusion Criteria.**

soft tape measure was then used to measure around the midpoint of the line between the lower rib margin and the highest point of the iliac crest. The WWI was calculated by dividing waist circumference (cm) by the square root of weight (kg) [14]. Total scores ranged from 0 to 24.29. We categorized these scores into two groups based on dichotomies, including lower (0–12.25) and higher (12.25–24.29) WWI.

## 2.3. Physical activity assessment

The CLHLS database assessed participants' physical activity situations through questionnaire responses. Respondents were asked to indicate their physical activity status based on the following categories: maintain physical activity, not physically active in the past but now physically active, previously physically active but no longer active, and never physically active. Each question had two possible responses: yes or no. Each question allowed for 'yes' or 'no' answers, with activities coded as dichotomous variables (participation or non-participation). We analyzed the 'yes' responses. Finally, physical exercise was coded as follows:0 if never performed, 1 if always performed, 2 if not performed in the past but currently performed, and 3 if performed in the past but not currently.

## 2.4. Multimorbidity assessment

Multimorbidity refers to the coexistence of two or more chronic diseases [3]. Through self-reported data, respondents were asked to specify whether they or their families had been diagnosed with any of 24 chronic conditions preceding death, including conditions like hypertension, diabetes, heart disease, stroke, bronchitis, tuberculosis, cataracts, glaucoma, cancer, prostate tumors, ulcers, Parkinson's, pressure sores, arthritis, dementia, epilepsy, gallbladder issues, dyslipidemia, rheumatism, chronic kidney disease, breast changes, uterine growths, prostate enlargement, and hepatitis. These were used to count the number of chronic diseases per participant, with two or more indicating multimorbidity. In this study, multimorbidity was defined based on diagnoses recorded in the database prior to or at the study index date (baseline). Participants who developed conditions during follow-up were not included in the primary analysis.

## 2.5. Mortality

Survival status and dates of death were obtained during the 2014 and 2018 follow-up visits. Dates of death were collected from officially issued death certificates or through interviews with next-of-kin or village doctors. Participants who were lost to follow-up before 2018 were considered censored values. The person years were calculated from 2011 (baseline) to the dates of death, loss to follow-up or the date of last interview in 2018, whichever came first.

## 2.6. Covariates

Covariates were derived from a standardized baseline questionnaire, covering [8] demographics (age, sex, ethnicity, marital status), socioeconomic status (education, place of birth, wealth, insurance) [15,16], and lifestyle behaviors (living pattern, smoking, alcohol consumption) [17,18]. Participants were categorized by ethnicity into Han Chinese and ethnic minority groups. Marital status was divided into married (with a surviving spouse) and others (separated, divorced, widowed, or never married). Wealth status was compared to others in the area and classified as "wealthy/above average" or "average/below." Living patterns included residing with household members, living alone, or being in an institutional setting. Participants were considered smokers/drinkers if they currently engaged in smoking/drinking, regardless of frequency and quantity.

## 2.7. Statistical analysis

Descriptive statistics summarized baseline characteristics of subjects, stratified by WWI. Continuous variables were reported as mean and SD, while categorical variables as percentages. The Cox proportional hazards model was used to

estimate HRs and 95% CIs, assessing the relationship between alterations in WWI and physical activity with all-cause mortality among patients with multimorbidity. Three models adjusted for potential confounders. Interaction and moderating effects were computed to examine the association between physical activity status and WWI. Stratified analyses using WWI investigated the combined impact of WWI and physical activity on all-cause mortality in patients with multimorbidity. Restricted cubic splines explored potential nonlinear relationships between WWI and all-cause mortality. Analyses were performed in Stata 17, SPSS 26, and R 4.4.1 for Windows. P values <0.05 were considered statistically significant.

## 3. Results

The baseline characteristics of 2,626 CLHLS participants are presented in Table 1. Nearly 39.40% of older adults had never been physically active. A total of 80.20% were males, 12.80% were born in urban areas, and 50.80% were in a normal marital status. Approximately 29.90% of respondents were current smokers and 25.40% were current drinkers.

Table 2 shows the impact of physical activity changes and WWI levels on all-cause mortality in multimorbidity patients, analyzed through three adjusted models. Consistent physical activity significantly reduced mortality (P<0.01). Model 1 demonstrated that persistently active patients exhibited a 41% lower mortality risk (HR: 0.59; 95% CI: 0.49–0.70). In comparison, individuals transitioning from inactive to active status experienced a more modest 17% reduction in mortality (HR: 0.83; 95% CI: 0.72–0.96).In contrast, those transitioning from active to inactive status showed a 26% increase in mortality (HR: 1.26; 95% CI: 1.06–1.48), while higher WWI levels were associated with a 13% elevated risk (HR: 1.13; 95% CI: 1.01–1.27). However, these associations attenuated following multivariable adjustment.

To further investigate the combined effect of physical activity and WWI on all-cause mortality in multimorbidity patients, we categorized WWI scores into two groups. Although no significant interaction or moderating effect between physical activity and WWI was found (see Tables S1 and S2), differential effects were observed across WWI levels (Table 3). In the higher WWI group, active patients had a stronger protective effect against all-cause mortality compared to inactive ones, with HRs of 0.540 (95% CI:0.423, 0.688) for maintain physical activity and 0.783 (95% CI:0.640, 0.958) for past inactivity followed by present exercise. Conversely, past but not current physical activity increased the risk (HR:1.265; 95% CI:1.009, 1.585). In the lower WWI group, physical activity statuses other than continuous activity had no significant impact on mortality. A forest plot visualizing these results is shown in Fig 2.

Furthermore, we stratified according to different physical exercise profiles and employed restricted cubic spline regression with four knots to investigate the potential nonlinear relationship between WWI and all-cause mortality in patients with multimorbidity across various physical activity scenarios, as depicted in Fig 3. We identified a significant nonlinear relationship between WWI and all-cause mortality in patients with multimorbidity, both overall and among those who were consistently physically active (p<0.01). As illustrated in Fig 3(A), we observed a significant increase in all-cause mortality for all patients with multimorbidity when the WWI exceeded 10.11. However, the all-cause mortality rate of patients with multimorbidity only exhibited a significant upward trend when their WWI exceeded 10.70 under conditions of continuous physical exercise, as depicted in Fig 3(B). This suggests that sustained physical exercise mitigates the impact of obesity on all-cause mortality in patients with multimorbidity. Conversely, nonlinear tests of the association between WWI and all-cause mortality in patients with multimorbidity did not yield significant results, whether considering a transition from inactivity to current physical activity, or conversely, from previous physical activity to present inactivity (p>0.1), as in Fig 3(C, D).

## 4. Discussion

In this prospective study of 2,626 participants with at least one multimorbidity from China's ageing cohorts, we report three key findings. First, physical activity reduces all-cause mortality in multimorbidity patients, benefiting both consistently active individuals and those who became active from a previously inactive state. Conversely, transitioning from active to inactive increases the risk of all-cause mortality. The potential mechanisms underlying this association may involve two pathways. Physical inactivity could directly contribute to metabolic dysfunction, systemic inflammation, and cardiovascular

**Table 1. Baseline characteristics of participants based on quartiles of WWI with complete data.**

| Characteristics | Total | Quintile 1 | Quintile 2 | Quintile 3 | Quintile 4 |
|---|---|---|---|---|---|
| Range of scores | 0-24.29 | 0-10.42 | 10.42-11.13 | 11.13-11.88 | 11.88-24.29 |
| N | 2626 | 662 | 651 | 658 | 655 |
| **Exercise (n, %)** | | | | | |
| Remain inactive | 1035 (39.4) | 275 (41.5) | 258 (39.6) | 233 (35.4) | 269 (41.1) |
| Remain active | 537 (20.4) | 118 (17.8) | 140 (21.5) | 155 (23.6) | 124 (18.9) |
| Inactive to active | 687 (26.2) | 176 (26.6) | 163 (25.0) | 182 (26.5) | 166 (24.2) |
| Active to inactive | 367 (14.0) | 93 (14.0) | 90 (13.8) | 88 (13.4) | 96 (14.7) |
| **Age, mean±SD** | 83.0 (10.0) | 82.3 (10.1) | 82.1 (9.9) | 82.4 (9.6) | 85.2 (10.0) |
| **Sex (n, %)** | | | | | |
| Females | 520 (19.8) | 117 (17.7) | 94 (14.4) | 94 (14.3) | 215 (32.8) |
| Males | 2106 (80.2) | 545 (82.3) | 557 (85.6) | 564 (85.7) | 440 (67.2) |
| **Ethnicity (n, %)** | | | | | |
| Han | 2497 (95.1) | 633 (95.6) | 620 (95.2) | 628 (95.4) | 616 (94.0) |
| Others (minority) | 129 (4.9) | 29 (4.4) | 31 (4.8) | 30 (4.6) | 39 (6.0) |
| **Education, mean±SD** | 3.6 (4.1) | 3.6 (4.1) | 4.0 (4.1) | 3.7 (4.1) | 3.2 (4.0) |
| **Place of birth (n, %)** | | | | | |
| Urban | 336 (12.8) | 74 (11.2) | 77 (11.8) | 106 (16.1) | 79 (12.1) |
| Rural | 2290 (87.2) | 588 (88.8) | 574 (88.2) | 552 (83.9) | 576 (87.9) |
| **Marital status (n, %)** | | | | | |
| Married (spouse alive) | 1335 (50.8) | 374 (56.5) | 346 (53.1) | 352 (53.5) | 263 (40.1) |
| Others | 1291 (49.2) | 288 (43.5) | 305 (46.9) | 306 (46.5) | 392 (59.8) |
| **Living pattern (n, %)** | | | | | |
| Living with family members | 2184 (83.2) | 559 (84.4) | 547 (84.0) | 560 (85.1) | 518 (79.1) |
| Living alone | 387 (14.7) | 93 (14.0) | 92 (14.1) | 82 (12.5) | 120 (18.3) |
| Living in an institution | 55 (2.1) | 10 (1.5) | 12 (1.8) | 16 (2.4) | 17 (2.6) |
| **Wealth (n, %)** | | | | | |
| Affluent | 538 (20.5) | 137 (20.7) | 129 (19.8) | 122 (18.5) | 150 (22.9) |
| Ordinary | 2088 (79.5) | 525 (79.3) | 522 (80.2) | 536 (81.5) | 505 (77.1) |
| **Smoking (n, %)** | | | | | |
| Ever | 789 (29.9) | 203 (30.7) | 212 (32.6) | 202 (30.7) | 167 (25.5) |
| Never | 1842 (70.1) | 459 (69.3) | 439 (67.4) | 456 (69.3) | 488 (74.5) |
| **Drinking (n, %)** | | | | | |
| Ever | 668 (25.4) | 158 (23.9) | 179 (27.5) | 187 (28.4) | 144 (22.0) |
| Never | 1958 (74.6) | 504 (76.1) | 472 (72.5) | 471 (71.6) | 511 (78.0) |
| **Insurance (n, %)** | | | | | |
| Medical payment insurance | 2101 (80.0) | 547 (82.6) | 521 (80.0) | 519 (78.9) | 514 (78.5) |
| Out of pocket | 525 (20.0) | 115 (17.4) | 130 (20.0) | 139 (21.1) | 141 (21.5) |

deterioration, consequently elevating mortality risk [19]. Alternatively, reverse causation may explain the observed pattern, wherein individuals with undiagnosed or progressive chronic conditions reduce activity levels following health status decline. Second, elevated levels of WWI are associated with an increased risk of all-cause mortality in multimorbidity patients. Third, stratified analysis shows that the impact of physical activity on all-cause mortality is more pronounced in multimorbidity patients with higher WWI levels compared to those with lower WWI levels. Specifically, both the protective and harmful effects of physical activity are more significant in the high WWI group.

Table 2. Association between different physical activity profiles and WWI exposure with all-cause mortality in patients with multimorbidity.

| Variables | Model1 | | Model2 | | Model3 | |
|---|---|---|---|---|---|---|
| | HR (95%CI) | P | HR (95%CI) | P | HR (95%CI) | P |
| Exercise | | | | | | |
| Remain inactive | Ref | | Ref | | Ref | |
| Remain active | 0.59 (0.49,0.70) | <0.001 | 0.65 (0.54,0.78) | <0.001 | 0.69 (0.57,0.83) | <0.001 |
| Inactive to active | 0.83 (0.72,0.96) | 0.013 | 0.95 (0.82,1.10) | 0.459 | 0.95 (0.82,1.10) | 0.527 |
| Active to inactive | 1.26 (1.06,1.48) | 0.007 | 1.10 (0.93,1.30) | 0.286 | 1.12 (0.94,1.32) | 0.201 |
| WWI | | | | | | |
| Lower WWI | Ref | | Ref | | Ref | |
| Higher WWI | 1.13 (1.01,1.27) | 0.032 | 1.06 (0.94,1.19) | 0.318 | 1.06 (0.94,1.19) | 0.323 |

HR: Hazard Ratio, CI: Confidence Interval.

Model1: Crude.

Model2: Adjust: age, sex, ethnic, marital, place of birth.

Model3: Adjust: age, sex, ethnic, marital, place of birth, living pattern, education, wealth, insurance, smoke, drink.

Table 3. Combined effect of WWI and different physical activity profiles on all-cause mortality in patients with multimorbidity.

| Variables | Lower WWI | | Higher WWI | |
|---|---|---|---|---|
| | HR (95%CI) | P | HR (95%CI) | P |
| Remain inactive | Ref | | Ref | |
| Remain active | 0.634 (0.492,0.817) | <0.001 | 0.540 (0.423,0.688) | <0.001 |
| Inactive to active | 0.881 (0.714,1.087) | 0.237 | 0.783 (0.640,0.958) | 0.017 |
| Active to inactive | 1.238 (0.973,1.575) | 0.082 | 1.265 (1.009,1.585) | 0.042 |

HR: Hazard Ratio, CI: Confidence Interval.

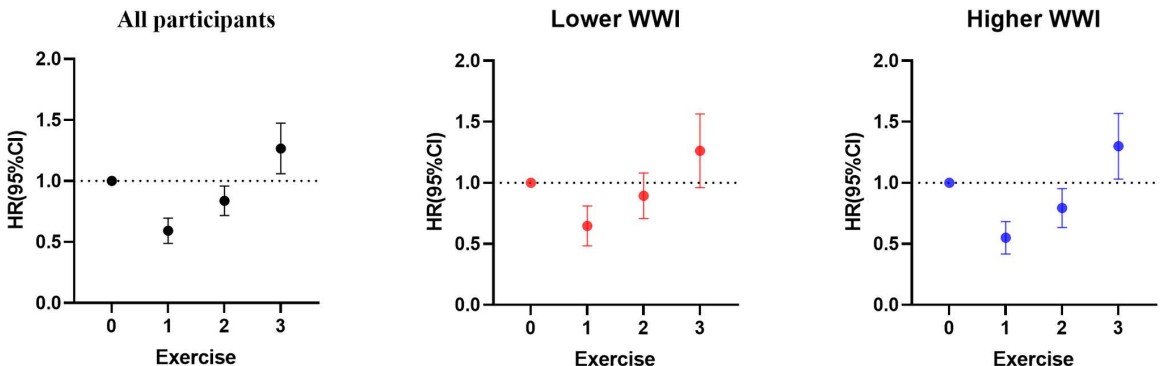

Fig 2. Hazard ratios (95% CI) of all-cause mortality in patients with multimorbidity by different physical activity profiles and stratified by the WWI. The three diagrams represent all participants, the lower WWI group, and the higher WWI group. The horizontal axis indicates physical activity levels:0 = no past or present activity, 1 = sustained activity from past to present, 2 = current activity with no past history, 3 = current but no past activity.

Multimorbidity is highly prevalent among older individuals and is frequently associated with an elevated risk of physical dysfunction and premature mortality [3,16]. As a result, epidemiological studies have extensively examined the relationship between physical activity and the prevalence of multimorbidity [20–22]. For example, a cohort study in a Chinese population found that moderate and high physical activity reduced multimorbidity risk [20]. Additionally, the Peking

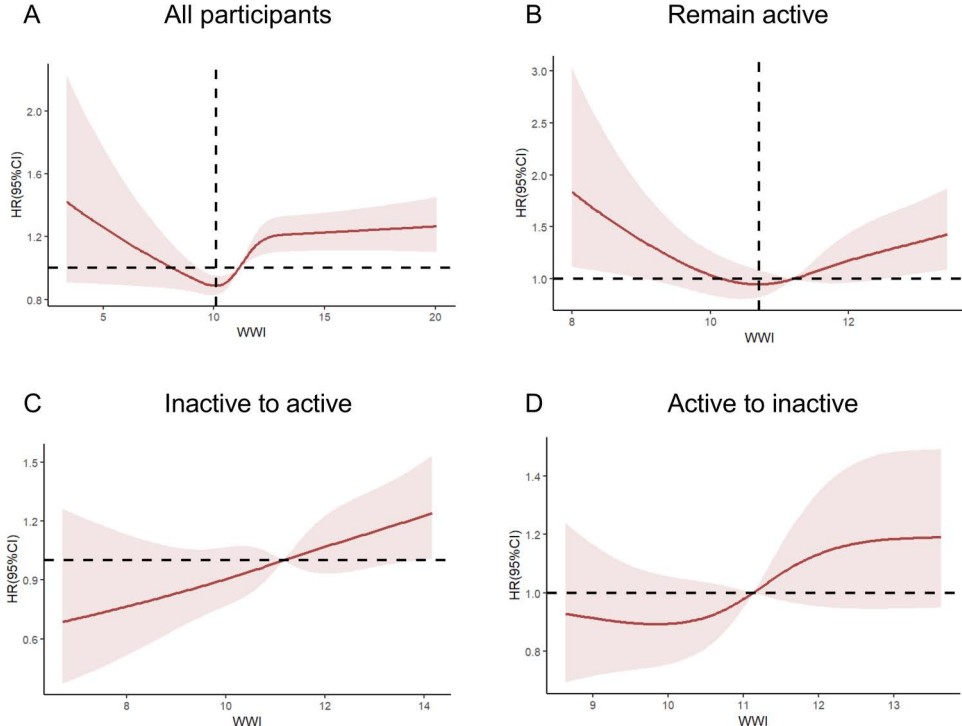

**Fig 3. Cubic splines for the WWI and all-cause mortality in patients with multimorbidity and stratified by the different physical activity profiles.** The shaded area represents 95% CIs. All participants (A), sustained activity from past to present (B), current activity with no past history (C), and current but no past activity (D).

University Lancet Commission on Healthy Aging in China reported that older adults' health was linked to their physical activity in adolescence and adulthood [21]. Similarly, another study in Chinese adults showed that less leisure-time physical activity increased multimorbidity risk [22]. These findings align with our study, but they do not categorize exercise status or examine the effects of past and present physical activity on multimorbidity.

Our systematic review of the existing literature revealed a large number of studies investigating the association between obesity and multimorbidity [23–27]. One such longitudinal cohort study involving 7,597 Chinese participants showed a significant association between obesity and increased risk of multimorbidity [25]. Another study of adult residents in northeastern China found that individuals with severe obesity had a higher prevalence of multimorbidity, and these diseases often exhibited complex clustering [26]. In addition, Jun Shan and his team conducted a meta-analysis of 969,130 patients, revealing a dose-dependent relationship between higher BMI and increased multimorbidity risk [27]. In these studies, body mass index (BMI) is typically used as the primary measure of obesity. However, it has been noted that BMI is not effective in distinguishing between different types of fat and the ratio of body fat to lean body mass [9]. One study has examined the relationship between WWI and multimorbidity in older Chinese adults [28]. As a newly proposed simple indicator of obesity, WWI can reflect both fat and muscle mass components [29]. However, this study showed obesity's link to higher multimorbidity prevalence but didn't explore its impact on all-cause mortality in multimorbidity patients.

Most studies have examined the relationship between physical activity or obesity and multimorbidity in isolation [8,28,30]. However, few studies have examined how WWI and different physical activity profiles together affect all-cause mortality in multimorbidity patients. We stratified WWI to compare the effects of four physical activity conditions on all-cause mortality at varying WWI levels. Our findings indicate that the impact of physical activity profiles on all-cause

mortality varies with WWI levels, being more significant at higher WWI levels. This implies that physical activity is particularly crucial for multimorbidity patients with high WWI. Additionally, based on the RCS results of this study, consistent physical activity reduces all-cause mortality in obese multimorbidity patients and mitigates obesity's impact on mortality. From a metabolic perspective, increased physical activity reduces serum triglycerides and arterial stiffness while enhancing mitochondrial respiratory efficiency, fat oxidation, and insulin sensitivity. It also decreases hepatic fat accumulation and HbA1c levels [31,32]. Collectively, these physiological improvements may mitigate the risk of cardiovascular disease, type 2 diabetes, and other chronic conditions in obese individuals, ultimately lowering mortality [33,34]. Consistent with our findings, Ying Chen et al.'s study on aerobic exercise and obesity suggests that exercise in obese patients can help prevent chronic diseases and mortality [35]. However, another study reports that physical activity is more effective in reducing the risk of death in frail older adults [36]. The discrepancy likely stems from sample selection limitations in their study. Our findings enhance understanding of the intricate links among anthropometric changes, varied physical activity patterns, and survival outcomes in multimorbidity patients.

In our study, stratified analyses showed a significant reduction in all-cause mortality among multimorbidity patients in the higher WWI group. However, further interaction and moderating effect analyses found no significant impact of WWI and physical activity on all-cause mortality. This may be due to the limitations of stratified analyses, which can reveal subgroup differences but may miss complex interactions. Additionally, WWI's specificity as an obesity measure might not fully reflect overall health or fitness. The varied effects of physical activity across populations and potential uncontrolled confounders could also influence the results. Future studies should use more rigorous methods and comprehensive data collection to explore the relationship between WWI, physical activity, and all-cause mortality.

Multimorbidity is a growing global health challenge, necessitating effective management strategies to improve patient outcomes. However, Current clinical care often suffers from lack of coordination, treatment duplication, excessive focus on single diseases, and polypharmacy [3]. To address this challenge, we suggest that the multimorbidity management paradigm should shift towards a person-centered approach. In this process, it is particularly important to assist patients in self-management [37]. For example, encouraging patients to maintain good exercise habits and weight control are measures that may contradict the lifestyle habits of patients with multimorbidity [38]. While self-management has the potential to reduce healthcare utilization, we must recognize the multifaceted benefits of physical activity for patients with multimorbidity [7]. Facing the challenge of multimorbidity, our findings underscore the need for tiered intervention strategies. Clinically, structured physical activity programs should be integrated into standard care protocols, while public health initiatives must prioritize age-tailored weight management support. This combined approach—pairing personalized clinical management with community-based prevention—represents the most viable pathway to sustained health improvements.

## 5. Strengths and limitations

The study has several strengths. First, weight and waist circumference measurements were assessed by trained researchers, reducing misclassification bias. Second, physical activity identification was accurate and comprehensive, covering past and present exercise conditions. Third, this study is one of the few to measure the relationship between changes in obesity metrics and physical activity on all-cause mortality in a multimorbidity population, providing valuable insights into the relationships among obesity, physical activity, and multimorbidity outcomes.

However, our study has limitations. First, in examining the combined effect of the WWI and physical activity profiles on all-cause mortality in multimorbidity patients, we could not account for certain potential confounders due to data limitations. These unadjusted covariates might affect the physical activity–mortality association, warranting further model optimization in future studies to address these factors. Second, Participants' self-reported physical activity may be subject to measurement bias, whereas their self-reported disease profiles and time of death could introduce recall bias in mortality analyses of comorbid patients. Moreover, the assessment of physical activity in this study was based on binary variables provided by CLHLS. While this dichotomous measure effectively captures basic participation status, the database does

not include parameters such as exercise frequency, intensity, or duration. Consequently, our analysis cannot account for potential variations across different activity levels. Third, follow-ups were conducted every two to four years, potentially missing temporal transitions between physical activity and WWI levels. Finally, the CLHLS database limits generalizability to other countries. Future studies should gather more detailed data to enhance understanding of the relationship between physical activity, WWI, and multimorbidity.

## 6. Conclusion

The findings indicate that sustained engagement in regular physical activity—including transitioning from inactivity to activity—significantly reduces mortality risk among older adults with multimorbidity. Conversely, higher WWI is independently associated with increased mortality in this population. Notably, both initiating physical activity and maintaining regular exercise were associated with reduced mortality risk in multimorbidity patients with higher WWI. These results strongly support integrating structured exercise interventions into clinical management protocols for this vulnerable group. We recommend routine WWI assessment and physical activity monitoring as part of geriatric evaluations, as this combined strategy may improve survival outcomes in multimorbid older adults.

## Supporting information

**S1 Table. Interaction analysis.**
(PDF)

**S2 Table. Moderating Effect.**
(PDF)

## Author contributions

**Conceptualization:** Kexin Ren.

**Data curation:** Bingbing Fan.

**Formal analysis:** Kexin Ren, Lang Li.

**Investigation:** Kexin Ren, Lang Li.

**Methodology:** Bingbing Fan.

**Software:** Bingbing Fan.

**Supervision:** Kexin Ren.

**Validation:** Kexin Ren, Lang Li.

**Visualization:** Kexin Ren.

**Writing – original draft:** Bingbing Fan.

**Writing – review & editing:** Kexin Ren.

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
