## [Decision Letter · Decision Letter 0]

22 Apr 2025

Dear Dr. Kexin Ren,

Thank you for submitting your manuscript to PLOS ONE. After careful consideration, we feel that it has merit but does not fully meet PLOS ONE’s publication criteria as it currently stands. Therefore, we invite you to submit a revised version of the manuscript that addresses the points raised during the review process.

** **

 The comments of the reviewer(s) are included at the bottom of this letter.

We look forward to receiving your revised manuscript.

Kind regards,

Maha Gasmi, PhD

Guest Editor

PLOS ONE

2. Please include captions for your Supporting Information files at the end of your manuscript, and update any in-text citations to match accordingly. Please see our Supporting Information guidelines for more information: http://journals.plos.org/plosone/s/supporting-information .

Editor Comments: 

1. Please check the manuscript for grammatical errors and awkward phrasing.

2. Material and method section: The exclusion of participants without multimorbidity prior to death is justified, but it would be helpful to provide more information on how "multimorbidity" was specifically defined or identified. Was it based solely on self-reports or was any medical documentation used to confirm diagnoses?

3. It would be useful to clarify whether participants who developed multimorbidity during the study were included or excluded.

4. The categorization of physical activity based on responses to the questionnaire seems to lack some nuance. For example, how were the activities coded as "yes" or "no"? Was it based on frequency, intensity, or duration of physical activity? Clarifying these details would improve the understanding of what "physical activity" means in the context of this study.

5. Discussion section: While the summary is effective, the section could provide a more detailed explanation of why these findings are important in the context of multimorbidity management. For example, while the study identifies the protective role of physical activity and increased risk of mortality due to WWI, a deeper exploration of how these results inform clinical practice or public health strategies for multimorbidity patients would be helpful.

Reviewer 1:

Giving a more convincing explanation of why the weight-adjusted waist index (WWI) should be used rather than more traditional metrics like BMI would improve the text.

Methodology:

Any explanations about the thresholds used should be included in the criteria for classifying WWI levels into "lower" and "higher" categories. This would improve reader comprehension and reproducibility.

It would be beneficial to have more information on the confounders that are adjusted for in the Cox proportional hazards models. Were lifestyle and socioeconomic factors fully controlled for?

The findings imply that regular exercise lessens the detrimental effects of high WWI mortality. It could be worthwhile to talk about potential explanations for this protective effect, such how exercise can lower inflammation or enhance metabolic health.

Limitations and Discussion:

Even if the study emphasizes the importance of physical exercise, it would be helpful to examine why changing from an active to an inactive state raises the risk of death. Could this be because people stopped exercising because their health deteriorated?

Recognize that using self-reported physical activity data may have drawbacks, such as recall bias.

Implications for Policy:

Based on these data, think about making specific recommendations for healthcare practitioners. How, for example, could recommendations for physical exercise be modified for senior citizens with high WWI and multimorbidity?

The forest plot visualization in Figure 2 effectively highlights the interaction effects. Ensure that the figure legends provide complete information for interpretation without referring back to the main text.

Reviewer 2

1. justification would be needed for the selection of this year and these respondents from 2011 to 2018 from the Chinese Longitudinal Healthy Longevity Survey (CLHLS)

2. The author points out that there are limitations in comparing the results with data from other countries (the CLHLS database limits generalizability to other countries.), so it would not be correct to refer to these data in the discussion.

3. Conclusion – more precise information would be needed in the conclusions.

4. It is necessary to describe the purpose of the research more precisely and provide a meaningful and reasoned answer in the conclusions, based on specific results

Reviewers' comments:

Reviewer's Responses to Questions

**Comments to the Author**

1. Is the manuscript technically sound, and do the data support the conclusions?

Reviewer #1: Yes

Reviewer #2: Partly

2. Has the statistical analysis been performed appropriately and rigorously?

Reviewer #1: Yes

Reviewer #2: Yes

3. Have the authors made all data underlying the findings in their manuscript fully available?

Reviewer #1: Yes

Reviewer #2: Yes

4. Is the manuscript presented in an intelligible fashion and written in standard English?

Reviewer #1: Yes

Reviewer #2: Yes

Reviewer #1: Justification for the Research:

Giving a more convincing explanation of why the weight-adjusted waist index (WWI) should be used rather than more traditional metrics like BMI would improve the text. The reader's comprehension may be improved by outlining the significance of World War I in forecasting health outcomes, particularly for elderly persons.

Methodology:

Any explanations about the thresholds used should be included in the criteria for classifying WWI levels into "lower" and "higher" categories. This would improve reader comprehension and reproducibility.

It would be beneficial to have more information on the confounders that are adjusted for in the Cox proportional hazards models. Were lifestyle and socioeconomic factors fully controlled for?

The findings imply that regular exercise lessens the detrimental effects of high WWI mortality. It could be worthwhile to talk about potential explanations for this protective effect, such how exercise can lower inflammation or enhance metabolic health.

Limitations and Discussion:

Even if the study emphasizes the importance of physical exercise, it would be helpful to examine why changing from an active to an inactive state raises the risk of death. Could this be because people stopped exercising because their health deteriorated?

Recognize that using self-reported physical activity data may have drawbacks, such as recall bias.

Implications for Policy:

Based on these data, think about making specific recommendations for healthcare practitioners. How, for example, could recommendations for physical exercise be modified for senior citizens with high WWI and multimorbidity?

The forest plot visualization in Figure 2 effectively highlights the interaction effects. Ensure that the figure legends provide complete information for interpretation without referring back to the main text.

Reviewer #2: 1. justification would be needed for the selection of this year and these respondents from 2011 to 2018 from the Chinese Longitudinal Healthy Longevity Survey (CLHLS)

2. The author points out that there are limitations in comparing the results with data from other countries (the CLHLS database limits generalizability to other countries.), so it would not be correct to refer to these data in the discussion.

3. Conclusion – more precise information would be needed in the conclusions.

4. It is necessary to describe the purpose of the research more precisely and provide a meaningful and reasoned answer in the conclusions, based on specific results

**Do you want your identity to be public for this peer review?** For information about this choice, including consent withdrawal, please see our Privacy Policy

Reviewer #1: No

Reviewer #2: No

---

## [Author Response · Author response to Decision Letter 1]

7 May 2025

Dear Editor and Reviewers,

I would like to express my sincere gratitude for the opportunity to revise and resubmit our manuscript, "The joint effect of weight-adjusted waist index and physical activity on all-cause mortality in Chinese elderly patients with multimorbidity: A Study Based on the CLHLS from 2011 to 2018", for consideration for publication in PLOS ONE. The insightful comments and suggestions from the reviewers have greatly contributed to enhancing the quality and clarity of our research. We have carefully addressed each point raised during the review process and have made substantial revisions as detailed below.

Editor:

1. Please check the manuscript for grammatical errors and awkward phrasing.

RESPONSE: Thank you for your valuable feedback. I have carefully revised the manuscript to address grammatical issues and improve phrasing. Please let me know if any additional modifications are needed. I will be glad to make further revisions to meet the journal’s standards.

2. Material and method section: The exclusion of participants without multimorbidity prior to death is justified, but it would be helpful to provide more information on how "multimorbidity" was specifically defined or identified. Was it based solely on self-reports or was any medical documentation used to confirm diagnoses?

RESPONSE: We are very grateful for your insightful comments. In response to your valuable suggestions, we have carefully revised the section on page 6 (lines 111–113) to clarify how multimorbidity in participants was identified. We believe that these improvements have greatly enhanced the scientific rigor and readability of the manuscript.

3. It would be useful to clarify whether participants who developed multimorbidity during the study were included or excluded.

RESPONSE: Thank you for your helpful suggestion. As requested, we have revised the manuscript (Page 7, Lines 118–121) to explicitly clarify that participants who developed comorbidities during the study period were excluded from the analysis. We believe this addition further strengthens the methodological transparency of our study.

4. The categorization of physical activity based on responses to the questionnaire seems to lack some nuance. For example, how were the activities coded as "yes" or "no"? Was it based on frequency, intensity, or duration of physical activity? Clarifying these details would improve the understanding of what "physical activity" means in the context of this study.

RESPONSE: We sincerely appreciate the reviewer's insightful comment. Indeed, the physical activity measure was limited to participants' binary (yes/no) self-reports of any activity engagement and did not capture specific parameters such as frequency or intensity. We fully acknowledge this limitation and have now added a discussion on its potential implications in the Limitations section (see lines 321-324 on page 17). Thank you for highlighting this important point.

5. Discussion section: While the summary is effective, the section could provide a more detailed explanation of why these findings are important in the context of multimorbidity management. For example, while the study identifies the protective role of physical activity and increased risk of mortality due to WWI, a deeper exploration of how these results inform clinical practice or public health strategies for multimorbidity patients would be helpful.

RESPONSE: We are deeply grateful for your meticulous review and valuable feedback. In response to your insightful comments, we have thoroughly revised the manuscript and incorporated specific recommendations for clinical practice and public health interventions grounded in our study findings. These additions aim to address the complex challenge of multimorbidity more comprehensively, as now presented on page 16 (lines 301-306). We appreciate your guidance in enhancing the practical relevance of our research.

Reviewer 1: 

1. Giving a more convincing explanation of why the weight-adjusted waist index (WWI) should be used rather than more traditional metrics like BMI would improve the text.

RESPONSE: We sincerely appreciate the reviewer's valuable comment regarding the choice of obesity metrics. In response, we have now included a more detailed discussion on the rationale and advantages of using the weight-adjusted waist circumference index (WWI) compared to conventional measures like BMI (see page 3, lines 61-66 and page 4, lines 69-72). We believe that this supplement makes the adoption of our research indicators more reasonable.

Methodology:

2. Any explanations about the thresholds used should be included in the criteria for classifying WWI levels into "lower" and "higher" categories. This would improve reader comprehension and reproducibility

RESPONSE: We sincerely thank the reviewers for their valuable suggestion to clarify the WWI thresholds. In response to this important comment, we have now added clear values for the use of dichotomization to classify WWI levels in the “Methods” section (page 6, lines 98-99). We believe this addition will greatly improve the clarity and reproducibility of our study. We are very grateful for this insightful suggestion, which helps strengthen our manuscript.

3. It would be beneficial to have more information on the confounders that are adjusted for in the Cox proportional hazards models. Were lifestyle and socioeconomic factors fully controlled for?

RESPONSE: We sincerely appreciate the reviewers' insightful comments regarding potential confounders in our Cox regression analysis. In the current study, we have included basic lifestyle and socioeconomic variables available in our database; however, we fully acknowledge that the covariates may not be exhaustive due to limitations in the dataset. We have explicitly addressed this as a study limitation in the revised manuscript (page 17, lines 315-319), and we agree that this represents an important area for improvement in future research. We are grateful for this constructive feedback, which will significantly inform our future work in this area.

4. The findings imply that regular exercise lessens the detrimental effects of high WWI mortality. It could be worthwhile to talk about potential explanations for this protective effect, such how exercise can lower inflammation or enhance metabolic health.

RESPONSE: We are deeply grateful for the reviewer's valuable suggestion regarding the physiological mechanisms linking physical activity to reduced mortality risk in obese patients. As recommended, we have incorporated this explanatory framework (see page 15, lines 271-276), which indeed enhances the biological plausibility of our findings. We truly appreciate this insightful guidance, as it has helped us improve the depth and rigor of our discussion.

Limitations and Discussion:

5. Even if the study emphasizes the importance of physical exercise, it would be helpful to examine why changing from an active to an inactive state raises the risk of death. Could this be because people stopped exercising because their health deteriorated?

RESPONSE: We sincerely appreciate your insightful suggestion regarding the potential mechanisms underlying the increased mortality risk observed in comorbid patients transitioning from active to inactive states. As you recommended, we have incorporated this valuable discussion into our manuscript (see pages13, lines 225-230). Your comment has significantly strengthened the depth and clinical relevance of our findings.

6. Recognize that using self-reported physical activity data may have drawbacks, such as recall bias.

RESPONSE: We sincerely appreciate the reviewer's valuable observation regarding the study limitations. As suggested, we have incorporated this important consideration into the revised manuscript (Page 17, Lines 319-321) to provide a more comprehensive discussion of our study's limitations.

Implications for Policy:

7. Based on these data, think about making specific recommendations for healthcare practitioners. How, for example, could recommendations for physical exercise be modified for senior citizens with high WWI and multimorbidity?

RESPONSE: We sincerely appreciate the reviewer's insightful comments regarding the gaps in our research discussion. As suggested, we have incorporated specific recommendations for healthcare practitioners into the revised manuscript (Page16, Lines 301-306) to enhance the clinical applicability of our findings.

8. The forest plot visualization in Figure 2 effectively highlights the interaction effects. Ensure that the figure legends provide complete information for interpretation without referring back to the main text.

RESPONSE: We are truly grateful for the reviewer's thoughtful feedback and constructive suggestions regarding Figure 2. We completely agree that the figure legends should contain sufficient information for independent interpretation. In revising the figure, we encountered some difficulty in presenting the lengthy names of the physical activity groups while maintaining both clarity and visual simplicity. After careful consideration, we opted to include the full group descriptions in the figure notes rather than on the X-axis, though we fully acknowledge this may not be the ideal solution. We would be extremely appreciative if the reviewer could kindly share any specific recommendations for improving this aspect of the visualization. Once again, we sincerely thank the reviewer for these invaluable comments, which have already helped enhance the quality of our manuscript.

Reviewer 2

1. Justification would be needed for the selection of this year and these respondents from 2011 to 2018 from the Chinese Longitudinal Healthy Longevity Survey (CLHLS)

RESPONSE: We sincerely appreciate the reviewer’s valuable suggestion regarding the choice of study period. We selected data from 2011–2018 because this time frame includes all necessary variables for our analysis, and 2018 represents the most recent year of available data in this database. We have now clarified this rationale in the revised manuscript (page 4, lines 80–82) and are grateful for the opportunity to improve our paper. If further clarification or modifications are needed, we would be happy to address them promptly.

2. The author points out that there are limitations in comparing the results with data from other countries (the CLHLS database limits generalizability to other countries.), so it would not be correct to refer to these data in the discussion.

RESPONSE: We sincerely appreciate your insightful comments regarding the limitations of the databases used in our study, which initially restricted our ability to incorporate foreign studies for comparative analysis. Following your suggestion, we have revised the Discussion to better contextualize the findings in light of this limitation. We greatly appreciate your feedback and hope the revised manuscript meets your approval (See lines 239-244, 247-254 and 278-279 of the discussion section).

3. Conclusion – more precise information would be needed in the conclusions.

RESPONSE: We sincerely appreciate your valuable comments regarding the conclusions of our study. We have carefully revised the manuscript according to your suggestions. Should any additional modifications be required, we would be most grateful for your further guidance and will promptly address all concerns (see page 18, lines 331–339).

4. It is necessary to describe the purpose of the research more precisely and provide a meaningful and reasoned answer in the conclusions, based on specific results

RESPONSE: We sincerely appreciate your valuable suggestions regarding our manuscript. We have carefully revised the research objectives and conclusions in the abstract and introduction based on your comments. (see page 1, lines 10–14, page 2, lines 27–230 and page 4, lines 75–77) Should further modifications be required, we would be most grateful for your additional guidance and will promptly implement any necessary improvements to meet your expectations.

---

## [Decision Letter · Decision Letter 1]

21 May 2025

The joint effect of weight-adjusted waist index and physical activity on all-cause mortality in Chinese elderly patients with multimorbidity: A Study Based on the CLHLS from 2011 to 2018.

PONE-D-24-49087R1

Dear Dr. Ren,

We’re pleased to inform you that your manuscript has been judged scientifically suitable for publication and will be formally accepted for publication once it meets all outstanding technical requirements.

Kind regards,

Maha Gasmi, PhD

Guest Editor

PLOS ONE

Additional Editor Comments (optional):

Thank you for your thorough revisions. You have addressed all the comments appropriately.

Reviewers' comments:

Reviewer's Responses to Questions

**Comments to the Author**

Reviewer #1: All comments have been addressed

Reviewer #2: All comments have been addressed

2. Is the manuscript technically sound, and do the data support the conclusions?

Reviewer #1: Yes

Reviewer #2: Yes

3. Has the statistical analysis been performed appropriately and rigorously?

Reviewer #1: Yes

Reviewer #2: Yes

4. Have the authors made all data underlying the findings in their manuscript fully available?

Reviewer #1: Yes

Reviewer #2: Yes

5. Is the manuscript presented in an intelligible fashion and written in standard English?

Reviewer #1: Yes

Reviewer #2: Yes

Reviewer #1: (No Response)

Reviewer #2: No other comments. Authors the authors have taken into account and corrected the inaccuracies previously expressed.

**Do you want your identity to be public for this peer review?** For information about this choice, including consent withdrawal, please see our Privacy Policy

Reviewer #1: No

Reviewer #2: **Yes: ** Agita ABELE

---

## [Editor Report · Acceptance letter]

PONE-D-24-49087R1

PLOS ONE

Dear Dr. Ren,

I'm pleased to inform you that your manuscript has been deemed suitable for publication in PLOS ONE. Congratulations! Your manuscript is now being handed over to our production team.

Kind regards,

on behalf of

Dr. Maha Gasmi

Guest Editor

PLOS ONE